# No evidence for increased transmissibility from recurrent mutations in SARS-CoV-2

Lucy van Dorp [1,5 ✉], Damien Richard[2,3,5], Cedric C. S. Tan [1], Liam P. Shaw [4], Mislav Acman[1] & François Balloux [1 ✉]

COVID-19 is caused by the coronavirus SARS-CoV-2, which jumped into the human population in late 2019 from a currently uncharacterised animal reservoir. Due to this recent association with humans, SARS-CoV-2 may not yet be fully adapted to its human host. This has led to speculations that SARS-CoV-2 may be evolving towards higher transmissibility. The most plausible mutations under putative natural selection are those which have emerged repeatedly and independently (homoplasies). Here, we formally test whether any homoplasies observed in SARS-CoV-2 to date are significantly associated with increased viral transmission. To do so, we develop a phylogenetic index to quantify the relative number of descendants in sister clades with and without a specific allele. We apply this index to a curated set of recurrent mutations identified within a dataset of 46,723 SARS-CoV-2 genomes isolated from patients worldwide. We do not identify a single recurrent mutation in this set convincingly associated with increased viral transmission. Instead, recurrent mutations currently in circulation appear to be evolutionary neutral and primarily induced by the human immune system via RNA editing, rather than being signatures of adaptation. At this stage we find no evidence for significantly more transmissible lineages of SARS-CoV-2 due to recurrent mutations.

---

[1] UCL Genetics Institute, University College London, London WC1E 6BT, UK. [2] Cirad, UMR PVBMT, F-97410 St Pierre, Réunion, France. [3] Université de la Réunion, UMR PVBMT, F-97490 St Denis, Réunion, France. [4] Nuffield Department of Medicine, John Radcliffe Hospital, University of Oxford, Oxford OX3 9DU, UK. [5] These authors contributed equally: Lucy van Dorp, Damien Richard. ✉email: lucy.dorp.12@ucl.ac.uk; f.balloux@ucl.ac.uk

Severe acute respiratory coronavirus syndrome 2 (SARS-CoV-2), the causative agent of coronavirus disease 2019 (COVID-19), is a positive single-stranded RNA virus that jumped into the human population towards the end of 2019[1–4] from a yet uncharacterised zoonotic reservoir[5]. Since then, the virus has gradually accumulated mutations leading to patterns of genomic diversity. These mutations can be used both to track the spread of the pandemic and to identify sites putatively under selection as SARS-CoV-2 potentially adapts to its new human host. Large-scale efforts from the research community during the ongoing COVID-19 pandemic have resulted in an unprecedented number of SARS-CoV-2 genome assemblies available for downstream analysis. To date (21 September 2020), the Global Initiative on Sharing All Influenza Data (GISAID)[6,7] repository has >70,000 complete high-quality genome assemblies available. This is being supplemented by increasing raw sequencing data available through the European Bioinformatics Institute and NCBI Short Read Archive, together with data released by specific genome consortiums, including COVID-19 Genomics UK (https://www.cogconsortium.uk/data/). Research groups around the world are continuously monitoring the genomic diversity of SARS-CoV-2, with a focus on the distribution and characterisation of emerging mutations.

Mutations within coronaviruses, and indeed all RNA viruses, can arrive as a result of three processes. First, mutations arise intrinsically as copying errors during viral replication, a process which may be reduced in SARS-CoV-2 relative to other RNA viruses, due to the fact that coronavirus polymerases include a proof-reading mechanism[8,9]. Second, genomic variability might arise as the result of recombination between two viral lineages co-infecting the same host[10]. Third, mutations can be induced by host RNA-editing systems, which form part of natural host immunity[11–13]. While population genetics theory states that the majority of mutations are expected to be neutral[14], some may be advantageous or deleterious to the virus. Mutations that are highly deleterious, such as those preventing virus host invasion, will be rapidly purged from the population; mutations that are only slightly deleterious may be retained, if only transiently. Conversely, neutral and in particular advantageous mutations can reach higher frequencies.

Mutations in SARS-CoV-2 have already been scored as putatively adaptive using a range of population genetics methods[1,15–21], and there have been suggestions that specific mutations are associated with increased transmission and/or virulence[15,18,21]. Early flagging of such adaptive mutations could arguably be useful to control the COVID-19 pandemic. However, distinguishing neutral mutations (whose frequencies have increased through demographic processes) from adaptive mutations (which directly increase the virus' transmission) can be difficult[22]. For this reason, the current most plausible candidate mutations under putative natural selection are those that have emerged repeatedly and independently within the global viral phylogeny. Such homoplasic sites may arise convergently as a result of the virus responding to adaptive pressures.

Previously, we identified and catalogued homoplasic sites across SARS-CoV-2 assemblies, of which approximately 200 could be considered as warranting further inspection following stringent filtering[1]. A logical next step is to test the potential impact of these and other more recently emerged homoplasies on transmission. For a virus, transmission can be considered as a proxy for overall fitness[23,24]. Any difference in transmissibility between variants can be estimated using the relative fraction of descendants produced by an ancestral genotype. While sampling biases could affect this estimate, we believe such an approach is warranted here for two reasons. First, the unprecedented and growing number of SARS-CoV-2 assemblies calls for the development of computationally fast methods that scale effectively with data sets. Second, and more importantly, the current genetic diversity of the SARS-CoV-2 population lacks strong structure at a global level due to the large number of independent introductions of the virus in most densely sampled countries[1]. This leads to the worldwide distribution of SARS-CoV-2 genetic diversity being fairly homogenous, thus minimising the risk that a homoplasic mutation could be deemed to provide a fitness advantage to its viral carrier simply because it is overrepresented, by chance, in regions of the world more conducive to transmission.

In this work, we make use of curated alignment comprising 46,723 SARS-CoV-2 assemblies to formally test whether any identified recurrent mutation is involved in altering viral fitness. We find that none of the recurrent SARS-CoV-2 mutations tested are associated with significantly increased viral transmission. Instead, recurrent mutations seem to be primarily induced by host immunity through RNA-editing mechanisms, and likely tend to be selectively neutral, with no or only negligible effects on virus transmissibility.

## Results

**Global diversity of SARS-CoV-2.** The global genetic diversity of 46,723 SARS-CoV-2 genome assemblies is presented as a maximum likelihood phylogenetic tree (Fig. 1a). No assemblies were found to deviate by >32 single-nucleotide polymorphisms (SNPs) from the reference genome, Wuhan-Hu-1, which is consistent with the relatively recent emergence of SARS-CoV-2 towards the latter portion of 2019[1–5]. We informally estimated the mutation rate over our alignment as $9.8 \times 10^{-4}$ substitutions per site per year, which is consistent with previous rates estimated for SARS-CoV-2[1–4] (Figs. S1 and S2). This rate also falls in line with those observed in other coronaviruses[25,26] and is fairly unremarkable relative to other positive single-stranded RNA viruses, which do not have a viral proof-reading mechanism[27,28].

Genetic diversity in the SARS-CoV-2 population remains moderate with a mean pairwise SNP difference across isolates of 8.4 (4.7–13.5, 95% confidence interval). This low number of mutations between any two viruses currently in circulation means that, to date, we believe SARS-CoV-2 can be considered as a single lineage, notwithstanding taxonomic efforts to categorise extant diversity into sublineages[29]. Our data set includes viruses sequenced from 99 countries (Fig. 1b and Supplementary Data 1), with a good temporal coverage (Supplementary Fig. 1b). While some countries are far more densely sampled than others (Fig. 1b), the emerging picture is that fairly limited geographic structure is observed in the viruses in circulation in any one region. All major clades in the global diversity of SARS-CoV-2 are represented in various regions of the world (Fig. 1a and Supplementary Fig. 3), and the genomic diversity of SARS-CoV-2 in circulation in different continents is fairly uniform (Fig. 1c and Supplementary Fig. 3).

**Distribution of recurrent mutations.** Across the alignment, we detected 12,706 variable positions, with an observed genome-wide ratio of non-synonymous to synonymous substitutions of 1.88 (calculated from Supplementary Data 2). Following masking of putatively artefactual sites and phylogeny reconstruction, we detected >5000 homoplasic positions (5710 and 5793, respectively using two different masking criteria), see 'Methods' and Supplementary Figs. 4 and 5 and Supplementary Data 3. Recurrent mutations may be detected as a result of recombination, for which we find no strong evidence in SARS-CoV-2 (Supplementary Fig. 6 and 7), or sequencing or genome assembly artefacts[30]. In line with our previous work (ref. [1]; see 'Methods'), we therefore applied two stringent filtering approaches to delineate sets of well-supported homoplasic sites, which present strong candidates to test for ongoing selection. This resulted in 398 and 411 homoplasic sites in the alignments, respectively (Supplementary Figs. 4 and 5 and Supplementary Data 3). The current

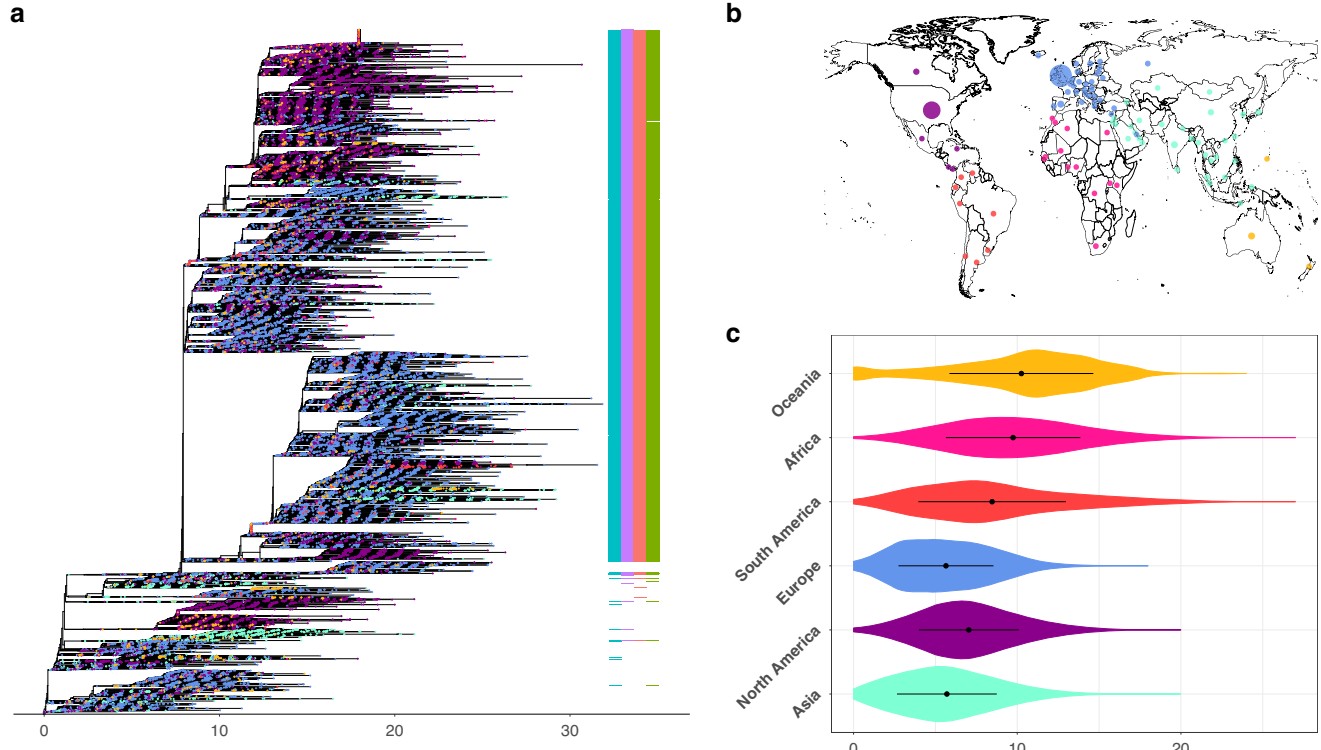

**Fig. 1 Overview of the global genomic diversity across 46,723 SARS-CoV-2 assemblies (sourced 30 July 2020) coloured as per continental regions.** **a** Maximum likelihood phylogeny for complete SARS-CoV-2 genomes. Tips are coloured by the continental region of sampling. D614G haplotype status is annotated by the presence/absence coloured columns (positions 241, 3037, 14,408 and 23,403, respectively). **b** Viral assemblies available from 99 countries displayed on a world map. **c** Within-continent pairwise genetic distance on a random subsample of 300 assemblies from each continental region. Colours in all three panels represent continents where isolates were collected. Magenta: Africa; Turquoise: Asia; Blue: Europe; Purple: North America; Yellow: Oceania; Dark Orange: South America according to metadata annotations available on GISAID (https://www.gisaid.org) and provided in Supplementary Data 1. The map in Fig. 1b was created using the R package rworldmap using the public domain Natural Earth data set.

distribution of genomic diversity across the alignment, together with identified homoplasic positions, is available as an open access and interactive web resource at: https://macman123. shinyapps.io/ugi-scov2-alignment-screen/.

As identified by previous studies[31–36], we find evidence of strong mutational biases across the SARS-CoV-2 genome, with a remarkably high proportion of C→U changes relative to other types of SNPs. This pattern was observed at both non-homoplasic and homoplasic sites (Supplementary Figs. 8–10). Additionally, mutations involving cytosines were almost exclusively C→U mutations (98%) and the distributions of $k$-mers for homoplasic sites appeared markedly different compared to that across all variable positions (Supplementary Figs. 11 and 12). In particular, we observed an enrichment in CCA and TCT 3-mers containing a variable base in their central position, which are known targets for the human APOBEC RNA-editing enzyme family[37].

**Signatures of transmission.** In order to test for an association between individual homoplasies and transmission, we defined a phylogenetic index designed to quantify the fraction of descendant progeny produced by any ancestral virion having acquired a particular mutation. We term this index the Ratio of Homoplasic Offspring (RoHO)[38]. In short, the RoHO index computes the ratio of the number of descendants in sister clades with and without a specific mutation over all independent emergences of a homoplasic allele (shown in red in Fig. 2). We confirmed that our approach is unbiased (i.e. produced symmetrically distributed RoHO index scores around the $\log_{10}(\text{RoHO}) = 0$ expectation for

recurrent mutations not associated with transmission) both by analysing simulated nucleotide alignments and discrete traits randomly assigned onto the global SARS-CoV-2 phylogeny (see 'Methods', Supplementary Fig. 13).

We restricted the analysis of the global SARS-CoV-2 phylogeny to homoplasies determined to have arisen at least $n = 3$ times independently. We observed 185 and 199 homoplasies passing all the RoHO score criteria under the more and less stringent masking procedures, respectively, and report in the main text the results obtained with the more stringent masking. We ignored all homoplasic events where the parent node led to fewer than two descendant tips carrying the ancestral allele and two with the derived allele (Fig. 2). In order to avoid pseudoreplication (i.e. scoring any genome more than once), we also discarded from the RoHO index calculations for any homoplasic parent node embedding a secondary homoplasic event involving the same site in the alignment (Fig. 2). Ignoring embedding homoplasic parent nodes led to only a marginal loss of statistical power and inclusion of homoplasies carried on embedded nodes yielded similar results (Supplementary Fig. 13b). Results were consistent for the alternative, less stringent, masking strategy (Supplementary Fig. 13c and Supplementary Data 4).

None of the 185 detected recurrent mutations having emerged independently a minimum of three times were statistically significantly associated with an increase in viral transmission for either tested alignment (paired $t$ test; Fig. 3, Supplementary Data 4 and Supplementary Fig. 13). We also did not identify any recurrent mutations statistically significantly associated with reduced viral transmissibility for the more stringently masked alignment. Instead,

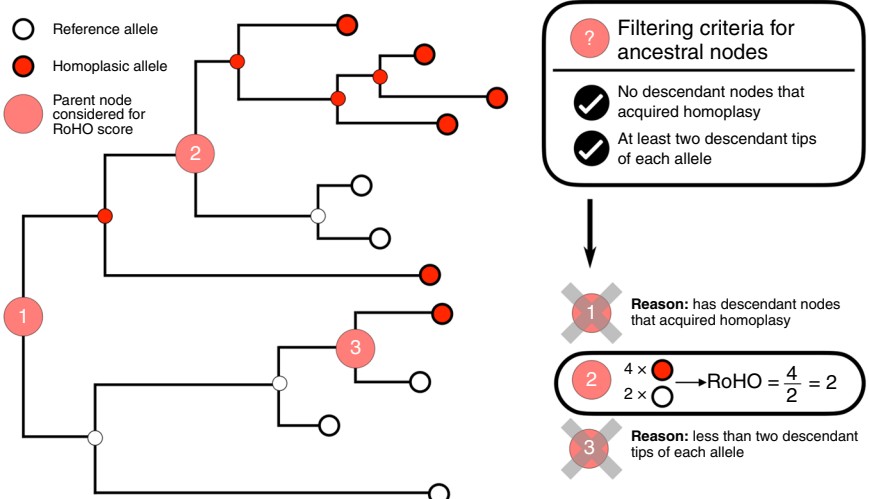

**Fig. 2 Schematic depicting the rationale behind the Ratio of Homoplasic Offspring (RoHO) score index.** White tips correspond to isolates carrying the reference allele and red tips to isolates carrying homoplasic alleles. This schematic phylogeny comprises three highlighted internal nodes annotated as corresponding to an ancestor who acquired a homoplasy. Node 3 is not considered because it fails our criterion of having at least two descendant tips carrying either allele. Node 1 is not considered because it includes embedded children nodes themselves annotated as carrying a homoplasic mutation. Node 2 meets our criteria: its RoHO score is $4/2 = 2$. In order to consider RoHO score for a homoplasic position, at least $n = 3$ nodes have to satisfy the criteria (not illustrated in the figure).

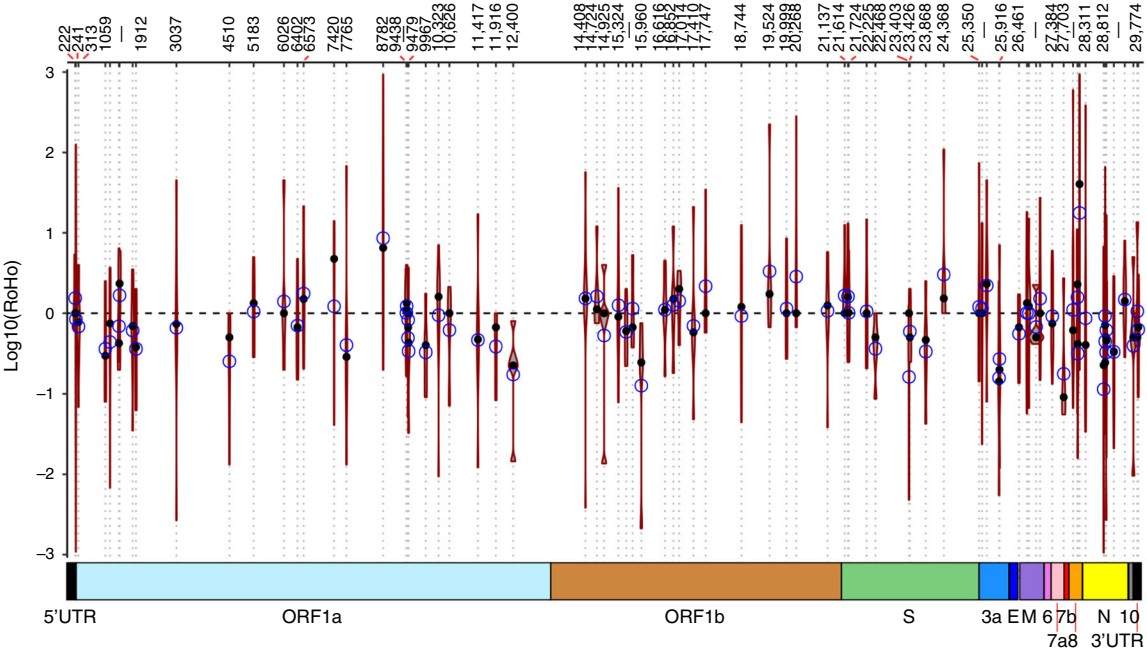

**Fig. 3 Genome-wide Ratio of Homoplasic Offspring (RoHO) values.** Confidence intervals show the $\log_{10}$(RoHO) index for homoplasies that arose in at least five filtered nodes in the Maximum likelihood phylogeny of 46,723 SARS-CoV-2 isolates. Black dot: median RoHO value; blue circle: mean RoHO value; error bar: standard deviation. Associated values including the number of replicates are provided in Supplementary Data 4 with the distribution for sites for which we have three replicates provided in Supplementary Fig. 13a. Top scale provides positions of the homoplasies relative to the Wuhan-Hu-1 reference genome and the bottom coloured boxes correspond to encoded ORFs. No homoplasy displayed a RoHO index distribution significantly different from zero (paired $t$ test, two-sided, alpha $= 0.05$). The number of replicates for each position is provided in Supplementary Data 4.

the entire set of 185 recurrent mutations seem to fit the expectation for neutral evolution with respect to transmissibility, with a mean and median overall $\log_{10}$RoHO score of $-0.001$ and $-0.02$, respectively. Moreover, the distribution of individual site-specific RoHO scores is symmetrically distributed around 0 with 97/185 mean positive values and 88/185 negative ones. To summarise, we would expect that recurrent mutations should be the best candidates for putative adaptation of SARS-CoV-2 to its novel human host.

However, none of the recurrent mutations in circulation to date shows evidence of being associated with viral transmissibility.

## Discussion

In this work, we analysed a data set of >46,700 SARS-CoV-2 assemblies sampled across 99 different countries and all major continental regions. Current patterns of genomic diversity highlight

multiple introductions in all continents (Fig. 1 and Supplementary Figs. 1–3) since the host-switch to humans in late 2019[1–4]. Across our data set, we identified a total of 12,706 mutations, heavily enriched in C→U transitions, of which we identified 398 strongly supported recurrent mutations (Supplementary Data 3 and Supplementary Figs. 4 and 5). Employing a phylogenetic index (RoHO) to test whether these recurrent mutations contribute to a change in transmission, we found no candidate convincingly associated with a significant increase or decrease in transmissibility (Figs. 2 and 3 and Supplementary Data 4).

Given the importance of monitoring potential changes in virus transmissibility, several other studies have investigated whether particular sets of mutations in SARS-CoV-2 are associated with changes in transmission and virulence[15,21,39]. We strongly caution that efforts to determine whether any specific mutation contributes to a change in viral phenotype, using solely genomic approaches, rely on the ability to distinguish between changes in allele frequency due to demographic or epidemiological processes, compared to those driven by selection[22]. A convenient and powerful alternative is to focus on sites that have emerged recurrently (homoplasies), as we do here. While such a method is obviously restricted to such recurrent mutations, it reduces the effect of demographic confounding problems, such as founder bias.

A much discussed mutation in the context of demographic confounding is D614G (nucleotide position 23,403), a non-synonymous change in the SARS-CoV-2 Spike protein. Korber et al. suggested that D614G increases transmissibility but with no measurable effect on patient infection outcome[21]. Other studies have suggested associations with increased infectivity in vitro[18,40] and antigenicity[41]. Here we conversely find that D614G does not associate with significantly increased viral transmission (median $\log_{10}(RoHO) = 0$, paired $t$ test $p = 0.28$; Supplementary Data 4), in line with our results for all other tested recurrent mutations. Though clearly, different choices of methodology may lead to different conclusions. A recent study on a sample of 25,000 whole-genome sequences exclusively from the UK used different approaches to investigate D614G. Not all analyses found a conclusive signal for D614G, and effects on transmission, when detected, appeared relatively moderate[39].

These apparently contrasting results for D614G should be considered carefully. What is, however, indisputable is that D614G emerged early in the pandemic and is now found at high frequency globally, with 36,347 assemblies in our data set (77.8%) carrying the derived allele (Fig. 1a and Supplementary Data 3). However, D614G is also in linkage disequilibrium (LD) with three other derived mutations (nucleotide positions 241, 3037, and 14,408) that have experienced highly similar expansions, as 98.9% of accessions with D614G also carry these derived alleles (35,954/36,347). It should be noted that the D614G mutation displays only five independent emergences that qualify for inclusion in our analyses (fewer than the other three sites it is associated with). While this limits our power to detect a statistically significant association with transmissibility, the low number of independent emergences suggests to us that the abundance of D614G is more probably a demographic artefact: D614G went up in frequency as the SARS-CoV-2 population expanded, largely due to a founder effect originating from one of the deepest branches in the global phylogeny, rather than being a driver of transmission itself.

The RoHO index developed here provides an intuitive metric to quantify the association between a given mutation and viral transmission. However, we acknowledge that this approach has some limitations. We have, for example, relied on admittedly arbitrary choices concerning the number of minimal observations and nodes required to conduct statistical testing. While it seems unlikely that this would change our overall conclusions, which are highly consistent for two tested alignments, results for particular

mutations should be considered in light of this caveat and may change as more genomes become available. Further, our approach necessarily entails some loss of information and therefore statistical power. This is because our motivation to test independent occurrences means that we do not handle "embedded homoplasies" explicitly: we simply discard them (Fig. 2), although inclusion of embedded homoplasies does not change the overall conclusions (Supplementary Fig. 13b). Finally, while our approach is undoubtedly more robust to demographic confounding (such as founder bias), it is impossible to completely remove all the sources of bias that come with the use of available public genomes.

In addition, it is of note that the SARS-CoV-2 population has only acquired moderate genetic diversity since its jump into the human population, and consequently, most branches in the phylogenetic tree are only supported by very few mutations. As a result of the low genetic diversity, most nodes in the tree have only low statistical support[42]. We therefore apply a series of stringent filters and masking strategies to the alignment (see 'Methods'). Also, while our method does not account quantitatively for phylogenetic uncertainty, we only computed RoHO scores for situations that should be phylogenetically robust (i.e. mutations represented in at least three replicate nodes, each with at least two representatives of the reference and alternate allele in descendants).

We further acknowledge that the number of SARS-CoV-2 genomes available at this stage of the pandemic, while extensive, still provides us only with moderate power to detect statistically significant associations with transmissibility for any individual recurrent mutation. The statistical power of the RoHO score methodology depends primarily on the number of independent homoplasic replicates rather than the strength of selection (Supplementary Fig. 14). The number of usable replicates per homoplasic site ranges between 3 and 14 and between 3 and 67 for the two masking strategies we applied (Supplementary Data 4). While the statistical power at most sites is weak, we predict a higher number of replicates at sites under strong positive selection, due to the expected recurrent mutations to the beneficial allelic state. We acknowledge that more sophisticated methods for phylodynamic modelling of viral fitness do exist[24,43,44]; however, these are not directly portable to SARS-CoV-2 and would be too computationally demanding for a data set of this size. Our approach, which is deliberately simple and makes minimal assumptions, is conversely highly scalable as the number of available SARS-CoV-2 genome sequences continues to rapidly increase.

To date, the fact that none of the 185 recurrent mutations in the SARS-CoV-2 population we identified as candidates for putative adaptation to its novel human host are statistically significantly associated with transmission suggests that the vast majority of mutations segregating at reasonable frequency are largely neutral in the context of transmission and viral fitness. This interpretation is supported by the essentially perfect spread of individual RoHO index scores around their expectation under neutral evolution (Fig. 3). However, it is nonetheless interesting to consider the cause of these mutations. Consistent with equivalent analyses (https://observablehq.com/@spond/linkage-disequilibrium-in-sars-cov-2, accessed 21 Sep 2020), we find no signature of recombination in SARS-CoV-2 (Supplementary Figs. 6 and 7), though 65% of the detected mutations comprise non-synonymous changes of which 38% derive from C→U transitions. This high compositional bias, as also detected in other studies[34–36], as well as in other members of the Coronaviridae[31–33], suggests that mutations observed in the SARS-CoV-2 genome are not solely the result of errors by the viral RNA polymerase during virus replication[35,36]. One possibility is the action of human RNA-editing systems, which have been implicated in innate and adaptive

immunity. These include the AID/APOBEC family of cytidine dea-minases, which catalyse deamination of cytidine to uridine in RNA or DNA, and the ADAR family of adenosine deaminases, which catalyse deamination of adenosine to inosine (recognised as a guanosine during translation) in RNA[45,46].

The exact targets of these host immune RNA-editing mechan-isms are not fully characterised but comprise viral nucleotide sequence target motifs whose editing may leave characteristic biases in the viral genome[37,47,48]. For example, detectable deple-tion of the preferred APOBEC3 target dinucleotide sequence TC has been reported in papillomaviruses[49]. In the context of SARS-CoV-2, Simmonds[36] and Di Giorgio et al.[35] both highlight the potential of APOBEC-mediated cytosine deamination as an underlying biological mechanism driving the over-representation of C→U mutations. However, APOBEC3 was shown to result in cytosine deamination but not hypermutation of HCoV-NL63 in vitro[50], which may suggest that additional biological processes also play a role.

In summary, our results do not point to any candidate recur-rent mutation significantly increasing transmissibility of SARS-CoV-2 at this stage and confirm that the genomic diversity of the global SARS-CoV-2 population is currently still very limited. It is to be expected that SARS-CoV-2 will diverge into phenotypically different lineages as it establishes itself as an endemic human pathogen. However, there is no a priori reason to believe that this process will lead to the emergence of any lineage with increased transmission ability in its human host.

## Methods

**Data acquisition**. We downloaded 48,454 SARS-CoV-2 assemblies were down-loaded from GISAID on 30/07/2020 selecting only those marked as 'complete', 'low coverage exclude' and 'high coverage only'. To this data set, all assemblies of total genome length <29,700 bp were removed, as were any with a fraction of 'N' nucleotides >5%. In addition, all animal isolate strains were removed, including those from bat, pangolin, mink, cat and tiger. All samples flagged by NextStrain as 'exclude' (https://github.com/nextstrain/ncov/blob/master/defaults/exclude.txt) as of 30/07/2020 were also removed. Twenty one further accessions were also filtered from our phylogenetic analyses as they appeared as major outliers following phylogenetic inference and application of TreeShrink[51] despite passing other fil-tering checks. This left 46,723 assemblies for downstream analysis. A full metadata table, list of acknowledgements and exclusions is provided in Supplementary Data 1.

**Multiple sequence alignment and maximum likelihood tree**. All 46,723 assemblies were aligned against the Wuhan-Hu-1 reference genome (GenBank NC_045512.2, GISAID EPI_ISL_402125) using MAFFT v7.471[52] implemented in the rapid phylodynamic alignment pipeline provided by Augur 6.3.0 (github.com/nextstrain/augur). This resulted in a 29,903 nucleotide alignment. As certain sites in the alignment have been flagged as putative sequencing errors (http://virological.org/t/issues-with-sars-cov-2-sequencing-data/473), we followed two separate masking strategies. The first masking strategy is designed to test the impact of the inclusion of putative sequencing errors in phylogenetic inference, masking several sites within the genome (n = 68) together with the first 55 and last 100 sites of the alignment (the list of sites flagged as 'mask' is available at https://github.com/W-L/ProblematicSites_SARS-CoV2/blob/master/problematic_sites_sarsCov2.vcf, acces-sed 30 Jul 2020)[30]. We also employed a less stringent approach, following the masking strategy employed by NextStrain, which masks only positions 18,529, 29,849, 29,851 and 29,853 as well as the first 130 and last 50 sites of the alignment. A complete list of masked positions is provided in Supplementary Data 5. This resulted in two masked alignments of 46,723 and 46,745 assemblies with 12,706 and 12,807 SNPs, respectively.

Subsequently, for both alignments, a maximum likelihood phylogenetic tree was built using IQ-TREE 2.1.0 COVID release (https://github.com/iqtree/iqtree2/releases/tag/v2.1.0) as the tree-building method[53]. The resulting phylogenies were viewed and annotated using ggtree v1.16.6[54] (Fig. 1 and Supplementary Fig. 1). Site numbering and genome structure are provided for available annotations (non-overlapping open reading frames (ORFs)) using Wuhan-Hu-1 (NC_045512.2) as reference.

**Recombination analysis**. In order to test for the presence of recombination, we performed a LD analysis considering whether the correlation between SNPs tends to disappear with an increase in the distance separating them on the genome. A classical approach to do so is the use of LD decay curves, which represent LD as a function of the distance separating each SNP pair. We calculated metrics of LD ($r^2$ and D') across all pair-wise combinations of variant sites using Tomahawk 0.7.0 (https://github.com/mklarqvist/tomahawk). The relationship between LD and distance yielded a regression coefficient of $3.56^{e-7}$ and a proportion of explained variance of $4.52^{e-4}$ (Supplementary Fig. 6). Following the approach presented at https://observablehq.com/@spond/linkage-disequilibrium-in-sars-cov-2 (accessed 21 Sep 2020), we tested the significance of the fitted $r^2$ by performing 1000 per-mutations of the genome coordinates, recomputing the regression each time. In all cases, the observed values fell within the null distribution providing no evidence of recombination in the SARS-CoV-2 alignment (Supplementary Fig. 7).

**Phylogenetic dating**. We informally estimated the substitution rate and time to the most recent common ancestor of both masked alignments by computing the root-to-tip temporal regression implemented in BactDating v1.0.1[55]. Both align-ments exhibit a significant correlation between the genetic distance from the root and the time of sample collection following 10,000 random permutations of sampling date (Supplementary Fig. 2).

**Homoplasy screen**. The resulting maximum likelihood trees were used, together with the input alignments, to rapidly identify recurrent mutations (homoplasies) using HomoplasyFinder v0.0.0.9[1,56]. HomoplasyFinder employs the method first described by Fitch[57], providing, for each site, the site-specific consistency index and the minimum number of changes invoked on the phylogenetic tree. All ambiguous sites in the alignment were set to 'N'. HomoplasyFinder identified a total of 5710 homoplasies, which were distributed over the SARS-CoV-2 genome (Supplemen-tary Fig. 4). For the less stringent masking of the alignment, HomoplasyFinder identified a total of 5793 homoplasies (Supplementary Fig. 5).

As previously described, we filtered both sets of identified homoplasies using a set of thresholds attempting to circumvent potential assembly/sequencing errors (filtering scripts are available at https://github.com/liampshaw/CoV-homoplasy-filtering and see ref. [1]). Here we only considered homoplasies present in >46 isolates (0.1% of isolates in the data set), where the number of submitting and originating laboratories of isolates with the homoplasy was >1 and displaying a third allele frequency <0.2 of that of the second allele frequency. This avoids us taking forward homoplasies that have only been identified in a single location as well as those putatively arising from low levels of recombination. This resulted in 398 filtered sites (411 following a less stringent masking procedure) of which 397 overlap. A full list of sites is provided for both alignments in Supplementary Data 3.

In addition, we considered an additional filtering criterion to identify homoplasic sites falling close to homopolymer regions, which may be more prone to sequencing error. We defined homopolymer regions as positions on the Wuhan-Hu-1 reference with at least four repeated nucleotides. While homopolymer regions can arise through meaningful biological mechanisms, for example polymerase slippage, such regions have also been implicated in increased error rates for both nanopore[58] and Illumina sequencing[59]. As such, homoplasies detected near these regions (±1 nt) could have arisen due to sequencing error rather than solely as a result of underlying biological mechanisms. If this were true, we would expect the proportion of homoplasic sites near these regions to be greater than that of homopolymeric positions across the entire genome. We tested this by identifying homopolymer regions using a custom python script (https://github.com/cednotsed/genome_homopolymer_counter) and performing a binomial test on the said proportions. A list of homopolymer regions across the genome is provided in Supplementary Data 6. Twenty five of the 398 (6.3%) filtered homoplasies were within ±1 nt of homopolymer regions and this proportion was significantly lower as compared to that of homopolymeric positions across the reference (9.7%; p = 0.0095). As such, we did not exclude homopolymer-associated homoplasies and suggest that these sites are likely to be biologically meaningful.

To determine whether systematic biases were introduced in our filtering steps, we performed a principal component analysis (PCA) on the unfiltered list of homoplasies obtained from HomoplasyFinder (n = 5710). The input space of the PCA included 11 variables, of which 8 were dummy-coded reference/variant nucleotides and a further 3 corresponded to the minimum number of changes on tree, SNP count and consistency index output by HomoplasyFinder. Visualisation of PCA projections (Supplementary Fig. 10a) suggested that there was no hidden structure introduced by our homoplasy filtering steps. The first two PCs accounted for 56% of the variance and were mostly loaded by the variables encoding the reference and variant nucleotides (Supplementary Fig. 10b).

**Annotation and characterisation of homoplasic sites**. All variable sites across the coding regions of the genomes were identified as synonymous or non-synonymous. This was done by retrieving the amino acid changes corresponding to all SNPs at these positions using a custom Biopython (v.1.76) script (https://github.com/cednotsed/nucleotide_to_AA_parser.git). The ORF coordinates used (including the ORF1ab ribosomal frameshift site) were obtained from the asso-ciated metadata according to Wuhan-Hu-1 (NC_045512.2).

To determine whether certain types of SNPs are overrepresented in homoplasic sites, we computed the base count ratios and cumulative frequencies of the different types of SNPs across all SARS-CoV-2 genomes at homoplasic and/or non-homoplasic sites (Supplementary Figs. 8 and 9). In addition, we identified the

sequence context of all variable positions in the genome (±1 and ±2 neighbouring bases from these positions) and computed the frequencies of the resultant 3-mers (Supplementary Fig. 11) and 5-mers (Supplementary Fig. 12).

**Quantifying pathogen fitness (transmission).** Under random sampling, we expect that any mutation, irrespective of how it arrives, that positively affects a pathogen's transmission fitness will be represented in proportionally more descendant nodes. As such, a pathogen's fitness can be expressed simply as the number of descendant nodes from the direct ancestor of the strain having acquired the mutation, relative to the number of descendants without the mutation (schematic Fig. 2). We define this as the RoHO index (full associated code available at https://github.com/DamienFr/RoHO)[38].

HomoplasyFinder[56] flags all nodes of a phylogeny corresponding to an ancestor that acquired a homoplasy. We only considered nodes with at least two descending tips carrying either allele and with no children node embedded carrying a subsequent mutation at the same site (see Fig. 2). For each such node in the tree, we counted the number of isolates of each allele and computed the RoHO index. We finally restricted our analysis to homoplasies having at least $n = 3$ individual RoHO indices (i.e. for which three independent lineages acquired the mutation). The latter allows us to consider only nodes for which we have multiple supported observations within the phylogeny, conveniently accounting for phylogenetic uncertainty. Paired $t$ tests were computed for each homoplasy to test whether RoHO indices were significantly different from zero. To validate the methodology, this analysis was carried out on data analysed using two different masking strategies (Fig. 3, Supplementary Fig. 13 and masked sites available in Supplementary Data 5). Full metadata associated with each tested site, including the number and associated countries of descendant offsprings, are provided in Supplementary Data 4.

**Assessing RoHO performance.** To assess the performance of our RoHO index, we performed a set of simulations designed to test the distribution of RoHO values under a neutral model.

We simulated a 10,000 nucleotide alignment comprising 1000 accessions using the rtree() simulator available in Ape v5.3[60] and genSeq from the R package PhyTools v0.7-2.0[61] using a single rate transition matrix multiplied by a rate of $6 \times 10^{-4}$ to approximately match that estimated in ref. [1]. This generated a 8236-SNP alignment, which was run through the tree-building and homoplasy detection algorithms described for the true data, identifying 3097 homoplasies (pre-filtering). Specifying a minimum of three replicates and at least two descendant tips of each allele, we obtained a set of RoHO scores none of which differed significantly from zero (Supplementary Fig. 13e).

In parallel, we tested for any bias in the RoHO scores when a set of randomly generated discrete traits were simulated onto the true maximum likelihood phylogeny. To do so, we employed the discrete character simulator rTraitDisc() available through Ape v5.3[60] specifying an equilibrium frequency of 1 (i.e. neutrality) and a normalised rate of 0.002 (after dividing branch lengths by the mean edge length). This rate value was manually chosen to approximately reproduce patterns of homoplasies similar to those observed for homoplasies in the actual phylogeny. Simulations were repeated for 100 random traits. Considering the discrete simulated traits as variant (putative homoplasic) sites, we again evaluated the RoHO indices (applying filters mentioned previously) for these 100 neutral traits. Following Bonferroni correction, no sites were deemed statistically significant (Supplementary Fig. 13d).

In all cases, to mitigate the introduction of bias we only considered homoplasies with nodes with at least two tips carrying either allele, in order to avoid $1/n$ and $n/1$ comparisons (see node 3 in Fig. 2). We further enforced a minimum number of three replicates (Fig. 3, Supplementary Fig. 13 and Supplementary Data 4). While we discarded homoplasies located on 'embedded nodes' to avoid pseudoreplication (see node 1 in Fig. 2), we note that including such sites has no impact on our results (Supplementary Fig. 11b).

In addition, we assessed the statistical power to detect significant deviations from neutrality of the RoHO index according to (i) the number of independent emergences of a homoplasy in the phylogeny and (ii) the imbalance between offspring number for each allele (i.e. fitness differential conferred by the carriage of the derived allele). To do so, we generated 1000 replicates for each combination of independent emergences (counts) of a homoplasy and corresponding fitness differential values using results from both masked alignments. For each replicate, we drew tips for the number of descended tips from the actual homoplasic parent nodes at our 185 candidate mutations sites under putative selection (all 185 pooled). We then probabilistically assigned a state to each tip according to an offspring imbalance (e.g. 10%). We drew replicates until we obtained 1000 for each combination comprising at least two alleles of each type. The proportion of significant paired $t$ tests for each combination of independent homoplasic parent nodes and fitness differential (10–80%) is presented as a heatmap (Supplementary Fig. 14).

The statistical power depends primarily on the number of independent emergences (i.e. homoplasic parent nodes) rather than the fitness differential (Supplementary Fig. 14, see 'Discussion'). Beneficial alleles have a far higher chance to increase their allele frequency upon introduction than deleterious ones, which are expected to be readily weeded out from the population. Thus we expect to observe a disproportionally higher number of independent homoplasic parent nodes for beneficial alleles. As such, the RoHO score index is inherently better suited to identify mutations associated with increased transmissibility relative to deleterious ones.

**Reporting summary.** Further information on research design is available in the Nature Research Reporting Summary linked to this article.

## Data availability

All genomic data analysed are available, on registration, from GISAID (https://www.gisaid.org). A full list of accessions used is provided in Supplementary Data 1 together with acknowledgement of all originating and submitting laboratories.

## Code availability

All codes used to generate the RoHO method presented in this manuscript are available at https://github.com/DamienFr/RoHO[38]. In addition, we provide links to the code used to conduct homoplasy filtering (https://github.com/liampshaw/CoV-homoplasy-filtering), per site annotations (https://github.com/cednotsed/nucleotide_to_AA_parser.git) and to assess the number of homopolymer regions in the SARS-CoV-2 genome (https://github.com/cednotsed/genome_homopolymer_counter).

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

## Acknowledgements

L.v.D. and F.B. acknowledge financial support from the Newton Fund UK-China NSFC initiative (grant MR/P007597/1) and the BBSRC (equipment grant BB/R01356X/1). L.v.D. is supported by a UCL Excellence Fellowship. D.R. is supported by an NIHR Precision AMR award. L.P.S. acknowledges funding from the Antimicrobial Resistance Cross-council Initiative supported by the seven UK research councils (grant NE/N019989/1). Computational analyses were performed on UCL Computer Science cluster and the South Green bioinformatics platform hosted on the CIRAD HPC cluster. We additionally wish to acknowledge the very large number of scientists in originating and submitting laboratories who have readily made available SARS-CoV-2 assemblies to the research community. We additionally wish to thank helpful comments on Twitter and bioRxiv, in particular from Harald Ringbauer, Palle Villesen, Sally Otto and Daniel Falush.

## Author contributions

L.v.D. and F.B. conceived and designed the study; L.v.D., M.A., D.R., L.P.S. and C.C.S.T. analysed data and performed computational analyses; L.v.D. and F.B. wrote the paper with inputs from all co-authors.

## Competing interests

The authors declare no competing interests.
