## [Peer Review File · Nature Communications]

Reviewers' Comments:

Reviewer #1:

Remarks to the Author:

This paper uses a fairly straightforward phylogenetic technique to demonstrate that no recurrent (i.e. homoplastic) mutations in SARS-CoV-2 are subject to adaptive evolution. While I can certainly believe the central result, and the authors make some good points, I do have a number of concerns:

1. There have a number of papers showing that there is no positive selection in SARS-CoV-2 (e.g. by David Robertson). As such, this is 'just' another paper saying the same thing. Hence, although I enjoyed the analysis, I didn't really learn anything new.
2. The homoplasy analysis is fine as far as it goes, but what the authors need to remember is that recombination also causes homoplasy and coronaviruses are expected to have frequent recombination and hence frequent homoplasy. Indeed, the occurrence of homoplasy has been used as a means to test the occurrence of recombination (<https://academic.oup.com/mbe/article/18/8/1425/993423>). To my mind, recombination could be commonplace in these data but hard to 'prove' because the sequences are so similar. The authors need to consider this.
3. Following on, one of the major issues of this study is that it is based on a single representation of evolutionary history: i.e. a single tree. So, how do they account for phylogenetic error? Surely they should integrate their analysis across multiple trees? And how do they account for the large amount of phylogenetic uncertainty in the SARS-CoV-2 phylogeny, with frequent polytomies? Alas, large parts of the SARS-CoV-2 phylogeny look nothing like the tree presented in Figure 2: there is a generic lack of resolution in the phylogeny.
4. I do not understand why they claim that SARS-CoV-2 is 'clonal' and this is not a word that is commonly used in studies of virus evolution. Given the occurrence of frequent, but perhaps undetectable, recombination SARS-CoV-2 may be anything but clonal. I would just delete this statement.
5. Perhaps I shouldn't be, but I am bothered by the suggesting that they are looking for mutations associated with 'transmissibility'. I get it, but it seems very unnecessary. Why not just say 'fitness'? This seems far safer.
6. D614G. I don't disagree with what the authors say about this mutation but they are missing the point a bit. There is no doubt that D614G *has* increased in frequency globally. That is is not a common recurrent mutation is therefore irrelevant - it has increased in frequency and we know it increases infectivity in the lab. Selectively advantageous mutations do not necessarily have to be recurrent ones: a mutation fixed once on one lineage could easily be hugely beneficial. Similarly, the noted linkage to other mutations also misses the point: doesn't this apply that at least one of these is selectively advantageous? Why would they all be neutral?

Reviewer #2:

Remarks to the Author:

In this article, van Dorp and colleagues present a large scale analysis of the potential impact of SARS-CoV-2 mutations on this virus transmissibility. For this, they focus on those mutations that arose multiple times independently (homoplasies), which they investigate using a novel, simple index. The Ratio of Homoplastic Offspring (RoHO) simply compares the number of descendants in sister lineages respectively carrying the ancestral or derived state at a given position across at least 3 replicates. They identified 80 recurrent mutations for which a RoHO index could be calculated and in all of these cases RoHO scores were compatible with undistinguishable transmissibility of the 2 variants.

General opinion

It is a well-written piece based on a very simple idea (meant positively). Given the unprecedented number of genomes generated during the pandemic and the real possibility that these genomes lead to data-informed decision making (as we have already seen), developing simple approaches that can provide us with a glimpse into the unfolding of the pandemic at all scales is very much valuable. This article proposes such a tool, specifically focused on one of the arguably most important phenotype of the virus, its transmissibility. This phenotype has rightfully attracted much attention. It is by nature the phenotype genome sampling and (typically Bayesian) phylodynamic analysis are tailored to investigate. Currently, phylodynamic analyses do not really allow for the analysis of very large datasets. In this context, the index proposed by the authors could be useful to generate hypotheses regarding associations between mutations and transmissibility, which could be investigated further with other tools.

Major remarks

- 1) I think the authors should at least produce simple metrics about the final 80 recurrent mutations that they deem amenable to meaningful RoHO score calculation. While many metrics could be interesting here (e.g. average count of descendant nodes, etc.), the number of replicates and offspring imbalance should absolutely appear here. In table S12, the authors present the results of simulations that show that, as expected, the power of their methods is strongly dependent on the number of replicates (and to a lesser extent offspring imbalance). A quick look at table S4 shows that 40/80 of the mutations only meet the minimum requirement of 3 replicates and only 2 mutations reach the number of replicates which would allow a modest 50ish% power assuming a very high imbalance (80%). The authors should present their 80 mutations in this context. A possibility could be that they produce a figure which would essentially consist in the heatmap of their simulation (maybe simplified so that only cells with a power >50% are coloured) in which they would coplot the 80 mutations (eg under the form of a mutation count per cell). It seems clear that, at the moment, the data is so that the detection power of the method is relatively low. In my view, it does not disqualify the test (many many more genomes will be produced in the months and years to come and therefore increase the power of this test) but this should be more explicitly shown in the results and discussed more openly (the authors have included a paragraph about limited power in the discussion but only vaguely link it to their data/analyses).
- 2) A bit in line with the previous comment, the authors could maybe try to run proper phylodynamic analyses, at least on their best sampled pairs of lineages to estimate transmission rates in sister clades (eg with BEAST2) and show that they are not statistically different (at the same time providing an example of a likely use of their method as a hypothesis generator).
- 3) I may have missed it but I think that the period of time during which the 2 sister lineages are detected is not taken into account? This could easily mislead the RoHO score for an individual replicate, eg if a highly transmissible lineage was to be totally suppressed while the sister lineage was not (totally imaginable given the strong heterogeneity of non pharmaceutical interventions) > it could then be that counts between the 2 lineages do not differ when transmissibility does. I understand that multiplicity of detection can be expected to correct such effects but I wondered whether the authors had also considered a simple normalization of their offspring count by the duration of a lineage detection?
- 4) This brings me to a last remark about heterogeneity. Many of the simplifying assumptions about the index are bound to homogeneity: that sampling was homogeneous, that sequencing effort was homogenous, that interventions were homogenous, etc. The authors are well aware of these limitations but do not elaborate much on them. I think it would be meaningful they do. And it'd also be interesting that they present and discuss descriptive statistics about some of these potential heterogeneities for their 80 recurrent mutations, eg geographic location (incl. something as simple as country of origin), etc. so that the reader (and they) can get an even better feeling about the limitations of their approach given the data.

Minor remarks

- The low genomic diversity of SARS-CoV-2 is not a result particularly bound to this analysis and has already been observed many many times. I would rephrase by linking this observation to the next sentence:

„Although SARS-CoV-2 currently only shows limited genomic and (presumably) phenotypic diversity, it can be expected that it will diverge into phenotypically different lineages as it establishes itself as an endemic human pathogen.“

- Phylogenetic uncertainty: the authors do not really discuss this point but it would make sense to mention it, especially given the low genomic variability. Did the authors estimate branch support with UFBoot or even better TBE? If so, it'd be great to somehow integrate this to the ms, as the reasoning line is mostly based on sistership and reciprocal monophyly.

- P10, paragraph 2: I do not totally understand the rationale for the second criterion.

- P12, paragraph 2: not clear whether this paragraph (which repeats things mentioned earlier) belongs here and how it articulates with its surroundings. Could it be a copy/paste error?

- Typos:

o P7 „than being a driver a of transmission“

o P7 „papillomaviruses“ rather than „Papillomavirus“

o P12 „RoHO index according to to“

Sébastien Calvignac-Spencer

Reviewer #3:

Remarks to the Author:

Dorp et al. work tackles the crucial and controversial question of whether specific mutations acquired by SARS-CoV-2 during the 8-months pandemic are, in fact, associated with increased transmissibility, as suggested by the recent paper of Korber et al. (Cell 182, 1–16, 2020) and, therefore, fixed by strong positive selection due to an increase in fitness of the mutated strains, or simply the result of neutral genetic drift and multiple founder events.

Their approach to the question is the intuitive expectation that increased transmission fitness will be reflected, in a phylogeny, by mutations under positive selection in ancestral nodes having proportionally more descendent nodes than neutral mutations. To test this hypothesis, the authors developed a Ratio of Homoplastic Offspring (RoHO) index that quantifies the proportion of descending tips carrying the ancestral mutation, and used Paired t-tests to decide whether RoHO indices are significantly different from zero (which, essentially, constitutes the null hypothesis of random genetic drift). Their results show that none of the 308 strongly supported recurrent mutations, including D614G, passed the test, leading to the conclusion that current empirical data provide no evidence for the emergence of specific mutations associated with increased SARS-CoV-2 transmissibility.

The topic of the manuscript is certainly of great interest and, in principle, I think the RoHO index is an elegant and simple measure to evaluate a scenario of enhanced transmissibility along a phylogeny. Unfortunately, I also found some technical and methodological problems in the actual implementation of Dorp et al. analysis that cast doubts on the actual validity of their results and conclusion, at least at this stage.

Major remarks

1. The entire calculation is based on the unstated assumption that the ML phylogeny used to reconstruct ancestral states and calculate the the RoHO index is an accurate depiction of the evolutionary relationships among the sequences in the data set. Unfortunately, this is far from obvious. Given the relatively low mutation rate of SARS-CoV-2 and the resulting low genetic heterogeneity, the signal in the tree is likely to be mostly star-like, with a large number of short branches with virtually no (bootstrap or otherwise) support. In other words, the authors cannot be

sure that the branching patterns they are seeing in the tree is accurate and there is no attempt to quantify such inaccuracy. The results could be a complete artifact of a poor phylogeny. On the other hand, unreliability of the phylogeny could have been easily addressed by including in the calculation of the RoHO index only those branches or clades in the phylogeny that are well supported by either bootstrapping or any of the many statistical tests to assess reliability of the branching pattern.

2. In addition, given the low level of genetic variation, just one or few mutations shared by a set of strains may be sufficient to cluster such strains within the same clade or appear related in the tree. By including in the alignment the 308 mutations putatively affecting transmissibility, the authors are biasing their phylogeny inference, since if those mutations emerged independently as a result, for example, of multiple founder events or convergent evolution, at least some of the strains carrying those mutations may artificially cluster together in the tree, which introduces a significant bias, not considered by the authors, that may alter the results.

3. Along the same line of remark #2, it is actually unclear how the result that no mutation has an RoHO index significantly different than zero can distinguish between a scenario a neutral genetic drift versus a scenario of convergent evolution (driven by strong positive selection).

4. The generalizability of the results is also questionable considering that country-specific data in GISAID at the beginning of April were affected by severe sampling bias, with a few countries (e.g. US and UK) overrepresented but still lacking sufficient temporal or phylogenetic signal for reliable phylogeny inference or molecular clock calibration (see Mavian et al. *JMIR Public Health Surveill* 6(2), e19170, 2020).

Thank you for your comments and the opportunity to review and resubmit the manuscript. Please find our full point-by-point response to all reviewers comments in blue. All significant changes in the main text have been highlighted in green.

REVIEWER COMMENTS

Reviewer #1 (Remarks to the Author):

This paper uses a fairly straightforward phylogenetic technique to demonstrate that no recurrent (i.e. homoplastic) mutations in SARS-CoV-2 are subject to adaptive evolution. While I can certainly believe the central result, and the authors make some good points, I do have a number of concerns:

1. There have a number of papers showing that there is no positive selection in SARS-CoV-2 (e.g. by David Robertson). As such, this is 'just' another paper saying the same thing. Hence, although I enjoyed the analysis, I didn't really learn anything new.

We are glad the reviewer enjoyed the analysis, although we politely disagree with the assessment that we do not present anything new. Not only do we outline a novel method, but our submitted work also provides one of the largest and most up to date formal assessments of the impact of mutations in SARS-CoV-2, which is clearly a rapidly changing situation meaning new analyses are constantly warranted. As an example, in our initial submission we considered 23,090 genomes, compared to the 15,537 currently considered by MacLean et al. *bioRxiv* (which we now cite).

Our resubmission considers an initial dataset of over double the size (48,454 initial genomes, recently downloaded from GISAID on 30/07/2020). In addition, we specifically focus on a carefully curated set of homoplasies, which represent some of the strongest candidates for ongoing selection. While other methods for rapid detection of selection rely on codon-based models, or Bayesian approaches with pre-defined priors, our method is essentially model-free with the advantage of also being readily interpretable and scalable as the number of SARS-CoV-2 genomes continues to increase.

2. The homoplasmy analysis is fine as far as it goes, but what the authors need to remember is that recombination also causes homoplasmy and coronaviruses are expected to have frequent recombination and hence frequent homoplasmy. Indeed, the occurrence of homoplasmy has been used as a means to test the occurrence of recombination (<https://academic.oup.com/mbe/article/18/8/1425/993423>). To my mind, recombination could be commonplace in these data but hard to 'prove' because the sequences are so similar. The authors need to consider this.

We are aware that recombination can also lead to homoplasies and mentioned this clearly in the second paragraph of the introduction. There is currently no evidence for recombination in SARS-CoV-2, although indeed this would be difficult to detect given the low levels of current genetic diversity, as the reviewer points out. We have therefore considered this point carefully. We believe our approach to assessing whether a mutation is associated to differences in transmission is in fact largely immune to moderate levels of recombination since it does not matter whether (homoplastic) mutations were introduced in given lineages by *de novo* mutation or recombination. Of note, linkage disequilibrium patterns across the SARS-CoV-2 genome do not point to the presence of extensive recombination (<https://observablehq.com/@spond/linkage-disequilibrium-in-sars-cov-2>).

3. Following on, one of the major issues of this study is that it is based on a single representation of evolutionary history: i.e. a single tree. So, how do they account for phylogenetic error? Surely they

should integrate their analysis across multiple trees? And how do they account for the large amount of phylogenetic uncertainty in the SARS-CoV-2 phylogeny, with frequent polytomies? Alas, large parts of the SARS-CoV-2 phylogeny look nothing like the tree presented in Figure 2: there is a generic lack of resolution in the phylogeny.

There is only limited genetic diversity in SARS-CoV-2 and its global population has been sampled to unprecedented depth. This creates a tree where most branches are supported by only one or two mutations. This renders traditional methods for statistical support such as bootstrap resampling inadequate as they rely on convergent signal by a large number of marker SNPs.

As such, rather than generating multiple phylogenies, we instead seek to account for possible uncertainties in our downstream analysis, for example by only computing a RoHO score for recurrent mutations for which we have at least three replicates (parent nodes) and at least two of each offspring carrying the reference and homoplastic allele. We apologise if this was not clear in the text. We now state in the methods “The latter allows us to consider only nodes for which we have multiple supported observations within the phylogeny”. We also add a paragraph to the discussion explicitly stating the challenge of phylogenetic uncertainty and how we account for it.

4. I do not understand why they claim that SARS-CoV-2 is ‘clonal’ and this is not a word that is commonly used in studies of virus evolution. Given the occurrence of frequent, but perhaps undetectable, recombination SARS-CoV-2 may be anything but clonal. I would just delete this statement.

We have now deleted ‘clonal’ from this sentence.

5. Perhaps I shouldn’t be, but I am bothered by the suggesting that they are looking for mutations associated with ‘transmissibility’. I get it, but it seems very unnecessary. Why not just say ‘fitness’? This seems far safer.

As stated in the Introduction, for a virus ‘transmissibility’ is a proxy for ‘fitness’, and we believe the two terms are fundamentally interchangeable in the context of this work. While we appreciate that either term could be suitable for a manuscript targeted narrowly to the scientific community, we believe it is far easier for the wider public to understand the concept of ‘transmissibility’ than ‘fitness’. We also worry that the use of ‘fitness’ may lead to mistaken associations with severity, so actually think ‘transmissibility’ is the safer option.

6. D614G. I don’t disagree with what the authors say about this mutation but they are missing the point a bit. There is no doubt that D614G *has* increased in frequency globally. That it is not a common recurrent mutation is therefore irrelevant - it has increased in frequency and we know it increases infectivity in the lab. Selectively advantageous mutations do not necessarily have to be recurrent ones: a mutation fixed once on one lineage could easily be hugely beneficial. Similarly, the noted linkage to other mutations also misses the point: doesn’t this apply that at least one of these is selectively advantageous? Why would they all be neutral?

There is clearly a possible scenario where a virus carrying the set of four mutations associated to the D614G haplotype was introduced into a region early on in the pandemic, subsequently rising to high frequency due to a founder effect and completely neutral evolution (genetic drift). Indeed, this scenario is also clearly posited by Korber et al. *Cell* 2020. who first proposed the relevance of D614G.

The fact that D614G occurs with a set of linked sites suggests this scenario as a strong possibility. We now explicitly highlight the close linkage of these sites in Figure 1a. Only one other of these sites

corresponds to a non-synonymous change (14408, RdRp P323L). At the same time, we are aware of an increasing body of literature suggesting D614G may be involved in a more infective phenotype *in vitro*. We cited such a case in our original submission and include further citations to more recently published work in the current version of the manuscript. We also cite recent genomics work on a UK dataset of SARS-CoV-2 which detects a marginal though inconclusive role of D614G.

In our new, larger analysis, we identify 23 homoplastic events associated with D614G corresponding to five testable parent nodes. Despite the increase in sample size, applying our RoHO statistic we find that D614G and associated sites continue to show no significant deviation from neutrality with respect to transmission. We provide careful assessment of this observation in the Discussion.

Reviewer #2 (Remarks to the Author):

In this article, van Dorp and colleagues present a large scale analysis of the potential impact of SARS-CoV-2 mutations on this virus transmissibility. For this, they focus on those mutations that arose multiple times independently (homoplasies), which they investigate using a novel, simple index. The Ratio of Homoplastic Offspring (RoHO) simply compares the number of descendants in sister lineages respectively carrying the ancestral or derived state at a given position across at least 3 replicates. They identified 80 recurrent mutations for which a RoHO index could be calculated and in all of these cases RoHO scores were compatible with undistinguishable transmissibility of the 2 variants.

General opinion

It is a well-written piece based on a very simple idea (meant positively). Given the unprecedented number of genomes generated during the pandemic and the real possibility that these genomes lead to data-informed decision making (as we have already seen), developing simple approaches that can provide us with a glimpse into the unfolding of the pandemic at all scales is very much valuable. This article proposes such a tool, specifically focused on one of the arguably most important phenotype of the virus, its transmissibility. This phenotype has rightfully attracted much attention. It is by nature the phenotype genome sampling and (typically Bayesian) phylodynamic analysis are tailored to investigate. Currently, phylodynamic analyses do not really allow for the analysis of very large datasets. In this context, the index proposed by the authors could be useful to generate hypotheses regarding associations between mutations and transmissibility, which could be investigated further with other tools.

We are pleased that the reviewer enjoyed our work and sees its value. At this point in the pandemic we believe analytical methods, particularly those newly devised, should be simple, transparent and computationally tractable so as to use all of the available genomic data being generated by laboratories around the world. We have now conducted a full reanalysis, more than doubling the dataset in our initial submission, demonstrating the scalability of our approach.

Major remarks

1) I think the authors should at least produce simple metrics about the final 80 recurrent mutations that they deem amenable to meaningful RoHO score calculation. While many metrics could be interesting here (e.g. average count of descendant nodes, etc.), the number of replicates and offspring imbalance should absolutely appear here.

Supplementary Table S4 provides associated metrics for the homoplasies tested, including the number of replicates. We provide the mean and median $\log_{10}(\text{RoHO})$ scores here and have now included two extra columns providing the average count of descendent nodes with and without the homoplastic allele together with the number of originating countries. We now consider 185 sites in our reanalysis and provide results using two differently masked alignments.

In table S12, the authors present the results of simulations that show that, as expected, the power of their methods is strongly dependent on the number of replicates (and to a lesser extent offspring imbalance). A quick look at table S4 shows that 40/80 of the mutations only meet the minimum requirement of 3 replicates and only 2 mutations reach the number of replicates which would allow a modest 50ish% power assuming a very high imbalance (80%). The authors should present their 80 mutations in this context. A possibility could be that they produce a figure which would essentially consist in the heatmap of their simulation (maybe simplified so that only cells with a power >50% are coloured) in which they would coplot the 80 mutations (eg under the form of a mutation count per cell). It seems clear that, at the moment, the data is so that the detection power of the method is relatively low. In my view, it does not disqualify the test (many many more genomes will be produced in the months and years to come and therefore increase the power of this test) but this should be more explicitly shown in the results and discussed more openly (the authors have included a paragraph about limited power in the discussion but only vaguely link it to their data/analyses).

Thank you for the suggestion. We have now altered the colour scale of the heatmap presented in Figure S12 and provide results for both masked alignments. There may have been a slight misunderstanding on the interpretation of the power analysis, for which we apologise: our initial submission was not clear enough in this respect.

The take-home message from the power analysis is that statistical power to detect positively selected sites is primarily a function of the number of independent homoplasies (qualifying for inclusion in the RoHo score), rather than the strength of selection. Positive natural selection is expected to generate multiple recurrent mutation events with ≥ 2 descendants carrying the derived state. Statistical power of $\sim 70\%$ and above is attained with ~ 30 homoplastic replicates.

We now report the range of homoplastic recurrences for each of the masking strategies in the main text. We also now include a paragraph clarifying these points and further discussing the limitations of our approach in the Results/Discussion, explicitly mentioning the results from our power analysis: *"We further acknowledge that the number of SARS-CoV-2 genomes available at this stage of the pandemic, whilst extensive, still provides us only with moderate power to detect statistically significant associations with transmissibility for any individual recurrent mutation (Figure S12). The statistical power of the RoHO score methodology depends primarily on the number of independent homoplastic replicates rather than the strength of selection (Figure S12). The number of usable replicates per homoplastic site ranges between 3-14, and 3-67 for the two masking strategies we applied (Table S4). While the statistical power at most sites is weak, we predict a higher number of replicates at sites under strong positive selection, due to the expected recurrent mutations to the beneficial allelic state."*

As noted by the reviewer, increases in the size of genomic datasets will continually increase the power of this approach. We now explicitly state this in the discussion: *"Our approach, which is deliberately simple and makes minimal assumptions, is conversely highly scalable as the number of available SARS-CoV-2 genome sequences continues to rapidly increase"*.

2) A bit in line with the previous comment, the authors could maybe try to run proper phylodynamic analyses, at least on their best sampled pairs of lineages to estimate transmission rates in sister clades (eg with BEAST2) and show that they are not statistically different (at the same time providing an example of a likely use of their method as a hypothesis generator).

Currently it is not tractable to run a formal phylodynamic analysis on this size of alignment and we wish to avoid subsetting the alignment which would greatly reduce our statistical power (see response above). It is also not clear which sister clades to select given we have no compelling significant hit. However, we consider the ratio of descendants as offering a direct proxy for transmission without the need for additional temporal calibration.

3) I may have missed it but I think that the period of time during which the 2 sister lineages are detected is not taken into account? This could easily mislead the RoHO score for an individual replicate, eg if a highly transmissible lineage was to be totally suppressed while the sister lineage was not (totally imaginable given the strong heterogeneity of non pharmaceutical interventions) > it could then be that counts between the 2 lineages do not differ when transmissibility does. I understand that multiplicity of detection can be expected to correct such effects but I wondered whether the authors had also considered a simple normalization of their offspring count by the duration of a lineage detection?

Given we consider the ratio of descendants as the indicator of transmission fitness, which in effect captures the survival time of a lineage we would not wish to use any normalisation by time which would in effect cancel out any signal of lineage persistence. However, we acknowledge that by neglecting any temporal information, we may not have taken advantage of the full information in the data. Thus we are grateful to the reviewer for this suggestion. This prompted us to run exploratory analyses substituting the number of descendants by 'persistence' (i.e. range of sampling dates). The global results remained largely unaffected (see Figure below), but resulted in increased variance, a possible bias. We decided not to report these results as they are far too preliminary and the approach would require considerable improvements and extensive validation before being considered for publication.

4) This brings me to a last remark about heterogeneity. Many of the simplifying assumptions about the index are bound to homogeneity: that sampling was homogeneous, that sequencing effort was homogenous, that interventions were homogenous, etc. The authors are well aware of these limitations but do not elaborate much on them. I think it would be meaningful they do. And it'd also be interesting that they present and discuss descriptive statistics about some of these potential heterogeneities for their 80 recurrent mutations, eg geographic location (incl. something as simple as country of origin), etc. so that the reader (and they) can get an even better feeling about the limitations of their approach given the data.

This is a reasonable concern, but we believe that the lack of homogeneity in sampling, sequencing effort, and interventions do not actually present a worrying limitation. This is because the extensive transmission of SARS-CoV-2 around the world, with many independent introductions in different geographic regions, means we are handling an effectively random sample, irrespective of sampling location (see Figure 1c and Figure S3). Thus, undoubtedly biased sampling (by location) is much less problematic than it is for other circumstances.

Our homoplasy filtering strategy does however discount homoplasies which are e.g. only observed in a single submitting/originating laboratory or a single country, to both catch putative sequencing/assembly artefacts but also to ensure that the homoplasies taken forward are represented by more than a single study and therefore location. We now state this clearly in the Methods: *“This avoids us taking forward homoplasies which have only been identified in a single location”*. We now also provide the ‘number of countries’ as a variable in Supplementary Table S4: the median number of countries a homoplasy has been observed in is 11 (range: 2-90).

Minor remarks

- The low genomic diversity of SARS-CoV-2 is not a result particularly bound to this analysis and has already been observed many many times. I would rephrase by linking this observation to the next sentence:

„Although SARS-CoV-2 currently only shows limited genomic and (presumably) phenotypic diversity, it can be expected that it will diverge into phenotypically different lineages as it establishes itself as an endemic human pathogen.“

We have reworded this sentence to 'confirm' rather than 'highlight' this observation; given the media hype around distinct or 'mutant' lineages of SARS-CoV-2 we feel it is important to reiterate this point at the end of the discussion.

- Phylogenetic uncertainty: the authors do not really discuss this point but it would make sense to mention it, especially given the low genomic variability. Did the authors estimate branch support with UFBoot or even better TBE? If so, it'd be great to somehow integrate this to the ms, as the reasoning line is mostly based on sistership and reciprocal monophyly.

We did not consider a bootstrapping procedure for reasons set out in response to Reviewer 1, point 3. Instead, to account for phylogenetic uncertainty, we only test recurrent mutations we believe are well supported by filtering for those with a minimum number of descendants and replicates across the tree.

We have clarified this in the text and added a paragraph to the discussion:

“In addition, it is of note that the SARS-CoV-2 population has only acquired moderate genetic diversity since its jump into the human population and, consequently, most branches in the phylogenetic tree are only supported by very few mutations. As a result of the low genetic diversity, most nodes in the tree have only low statistical support [52]. This prompted us to apply a series of stringent filters and masking strategies to the alignment (see Methods). Also, while our method does not account quantitatively for phylogenetic uncertainty, we only computed RoHO scores for situations which should be phylogenetically robust (i.e. mutations represented in at least three replicate nodes, each with at least two representatives of the reference and alternate allele in descendants).”

- P10, paragraph 2: I do not totally understand the rationale for the second criterion.

This criterion was taken forward from our previous paper (van Dorp *et al.* (2020) *IGE*, 83:104351) identifying homoplasies: the idea was to only consider homoplasies for which a certain proportion of neighbours in the tree also carry the homoplasy (e.g. to avoid considering singleton cases which may be more likely to represent sequencing artefacts). However, this filter is now largely redundant because of both increased data, and because we only take forward and test sites for which we observe at least two of each descendent offspring (in our previous paper, all such sites met this second criterion). We have therefore removed it.

- P12, paragraph 2: not clear whether this paragraph (which repeats things mentioned earlier) belongs here and how it articulates with its surroundings. Could it be a copy/paste error?

Apologies, we have edited this paragraph to include reference to the examples for each exclusion criteria presented in Figure 2 and reorganised the discussion.

- Typos:

- o P7 „than being a driver a of transmission“
- o P7 „papillomaviruses“ rather than „Papillomavirus“
- o P12 „RoHO index according to to“

Corrected.

Sébastien Calvignac-Spencer

Reviewer #3 (Remarks to the Author):

Dorp et al. work tackles the crucial and controversial question of whether specific mutations acquired by SARS-CoV-2 during the 8-months pandemic are, in fact, associated with increased transmissibility, as suggested by the recent paper of Korber et al. (Cell 182, 1–16, 2020) and, therefore, fixed by strong positive selection due to an increase in fitness of the mutated strains, or simply the result of neutral genetic drift and multiple founder events.

Their approach to the question is the intuitive expectation that increased transmission fitness will be reflected, in a phylogeny, by mutations under positive selection in ancestral nodes having proportionally more descendent nodes than neutral mutations. To test this hypothesis, the authors developed a Ratio of Homoplastic Offspring (RoHO) index that quantifies the proportion of descending tips carrying the ancestral mutation, and used Paired t-tests to decide whether RoHO indices are significantly different from zero (which, essentially, constitutes the null hypothesis of random genetic drift). Their results show that none of the 308 strongly supported recurrent mutations, including D614G, passed the test, leading to the conclusion that current empirical data provide no evidence for the emergence of specific mutations associated with increased SARS-CoV-2 transmissibility.

The topic of the manuscript is certainly of great interest and, in principle, I think the RoHO index is an elegant and simple measure to evaluate a scenario of enhanced transmissibility along a phylogeny. Unfortunately, I also found some technical and methodological problems in the actual implementation of Dorp et al. analysis that cast doubts on the actual validity of their results and conclusion, at least at this stage.

We are pleased the reviewer found our approach intuitive and timely. We hope our point-by-point response and extended reanalysis demonstrates the scalability of our method and validity of our results and conclusions.

Major remarks

1. The entire calculation is based on the unstated assumption that the ML phylogeny used to reconstruct ancestral states and calculate the the RoHO index is an accurate depiction of the evolutionary relationships among the sequences in the data set. Unfortunately, this is far from obvious. Given the relatively low mutation rate of SARS-CoV-2 and the resulting low genetic heterogeneity, the signal in the tree is likely to be mostly star-like, with a large number of short branches with virtually no (bootstrap or otherwise) support. In other words, the authors cannot be sure that the branching patterns they are seeing in the tree is accurate and there is no attempt to quantify such inaccuracy. The results could be a complete artifact of a poor phylogeny. On the other hand, unreliability of the phylogeny could have been easily addressed by including in the calculation of the RoHO index only those branches or clades in the phylogeny that are well supported by either bootstrapping or any of the many statistical tests to assess reliability of the branching pattern.

We respectfully disagree that we made no attempt to quantify this inaccuracy. We agree that the phylogeny of SARS-CoV-2 is poorly resolved with traditional ML methods due to the low diversity of viruses currently in circulation; however, this is a problem which cannot be rectified through bootstrap analysis (see also response to reviewer 1 and 2).

Therefore, the careful set of filtering criteria we apply to both our candidate list of homoplasies and those recurrent mutations for which we compute a RoHO score is precisely designed to only take forward those sites which are the strongest candidates and the most phylogenetically robust (i.e. we only consider positions for which we observe at least three replicate nodes and for which we have at least two representatives of the reference and alternate allele in descendants). We now explicitly state this in the methods: *“The latter allows us to consider only nodes for which we have multiple supported observations within the phylogeny.”* While we can see how this may not be the quantification of inaccuracy the reviewer has in mind, we think requiring multiple independent branching patterns is a reasonable approach where bootstrapping is inappropriate.

In light of these comments we appreciate that our wording around this section placed a strong emphasis on identifying candidate sequencing errors, without also mentioning the motivation to reduce any impact of poorly resolved nodes in the phylogenetic tree. We now add a clear paragraph to this effect in the discussion:

“In addition, it is of note that the SARS-CoV-2 population has only acquired moderate genetic diversity since its jump into the human population and, consequently, most branches in the phylogenetic tree are only supported by very few mutations. As a result of the low genetic diversity, most nodes in the tree have only low statistical support [52]. This prompted us to apply a series of stringent filters and masking strategies to the alignment (see Methods). Also, while our method does not account quantitatively for phylogenetic uncertainty, we only computed RoHO scores for situations which should be phylogenetically robust (i.e. mutations represented in at least three replicate nodes, each with at least two representatives of the reference and alternate allele in descendants).”

2. In addition, given the low level of genetic variation, just one or few mutations shared by a set of strains may be sufficient to cluster such strains within the same clade or appear related in the tree. By including in the alignment the 308 mutations putatively affecting transmissibility, the authors are biasing their phylogeny inference, since if those mutations emerged independently as a result, for example, of multiple founder events or convergent evolution, at least some of the strains carrying those mutations may artificially cluster together in the tree, which introduces a significant bias, not considered by the authors, that may alter the results.

We are aware that a small number of erroneous mutations may lead to errors in the phylogenetic tree. We therefore conducted our analyses using two sets of stringent masking criteria to mitigate this effect. While our initial submission used the masking proposed by NextStrain (see Supplementary Table S5), we now present as main text figures results using the much more stringent masking under active maintenance by de Maio et al (last updated 30/07/2020). Results were highly consistent in both cases and are presented in full in Supplementary Tables S3 and S4.

We do not remove candidate mutations when building the phylogenetic tree given these are the specific sites we wish to test, using the phylogeny to do so.

3. Along the same line of remark #2, it is actually unclear how the result that no mutation has an RoHO index significantly different than zero can distinguish between a scenario a neutral genetic drift versus a scenario of convergent evolution (driven by strong positive selection).

First, we show that the log₁₀ of the RoHO index is 0 following simulations under neutral evolution (Figure S11d, S11e). Second, in Supplementary Figure S12 we provide a power analysis showing our ability to detect positive selection, using two differently processed alignments, given simulated imbalance in the transmission fitness of viruses carrying the mutation in question. These two analyses combined suggest that the RoHO index can distinguish these scenarios with appropriate power under reasonable assumptions.

4. The generalizability of the results is also questionable considering that country-specific data in GISAID at the beginning of April were affected by severe sampling bias, with a few countries (e.g. US and UK) overrepresented but still lacking sufficient temporal or phylogenetic signal for reliable phylogeny inference or molecular clock calibration (see Mavian et al. JMIR Public Health Surveill 6(2), e19170, 2020).

The situation has changed considerably since the beginning of April, and these sampling biases have reduced. The dataset has very strong temporal signal as shown in Figure S2. Furthermore, as stated above in our reply to reviewer 2, even though there are still many more sequences available from the UK, USA and Australia, than other continental regions, this has no impact on our characterisation of the global dataset due to the highly homogeneous distribution of diversity. Our filtering criteria also means we do not test any homoplastic site observed in a single submitting or originating laboratory, which to some extent avoids us considering mutations only observed in a single region/location. We now state this for clarity in the methods: *“This avoids us taking forward homoplasies which have only been identified in a single location”*. We now also provide the ‘number of countries’ as a variable in Supplementary Table S4: the median number of countries a homoplasy has been observed in is 11 (range: 2-90). Given the lack of geographic structure we would argue that our results are highly generalizable.

Reviewers' Comments:

Reviewer #1:

Remarks to the Author:

Unfortunately, the revised version of this paper, the responses to my comments, has done nothing to change my opinion here:

1. The authors must account for phylogenetic uncertainty and their defence against this is lacking. They could easily integrate over a posterior set of trees from MrBayes.
2. They have effectively ignored the point that homoplasies can be generated by recombination and that undermines their analysis: a homoplasy generated by recombination is obviously (!) not a recurrent mutation and its appearance would mean there is no evidence for a selective advantage.
3. The D614G mutation story is more complex than they say. The Volz paper in fact provides consistent evidence for a selective advance in this mutation. Hence, Van Dorp et al. are misrepresenting it. This highlights a major limitation of the whole Van Dorp study: it is not just recurrent mutations that will increase fitness. Indeed, a selective sweep on a single mutation is just as important as D614G shows. Hence, the whole focus on recurrent mutations is very limiting.

Reviewer #2:

Remarks to the Author:

I am generally satisfied by the clarifications, additional analyses and textual amendments presented by the authors. In particular, the limitations of the RoHo index now appear more clearly in the article.
Sebastien Calvignac-Spencer

Reviewer #3:

Remarks to the Author:

The authors did an excellent job addressing the reviewers' concerns. In particular, the new manuscript includes the analysis of a much larger data set, leading to the same results as before, and clearly discusses the potential problems linked to phylogenetic uncertainty. The filtering applied seems reasonable and strengthens the conclusions of the paper. I have no further comments. Very interesting work!

We have provided further edits to the manuscript highlighted in yellow with our previous amendments shown in green.

Reviewer #1 (Remarks to the Author):

Unfortunately, the revised version of this paper, the responses to my comments, has done nothing to change my opinion here:

1. The authors must account for phylogenetic uncertainty and their defence against this is lacking. They could easily integrate over a posterior set of trees from MrBayes.

We politely disagree with the reviewer. We clarified our reasoning and set out clearly how we do account for phylogenetic uncertainty in our response to all three reviewers and in the resubmission. For example, we stated to this reviewer in our last response:

“There is only limited genetic diversity in SARS-CoV-2 and its global population has been sampled to unprecedented depth. This creates a tree where most branches are supported by only one or two mutations. This renders traditional methods for statistical support such as bootstrap resampling inadequate as they rely on convergent signal by a large number of marker SNPs.

As such, rather than generating multiple phylogenies, we instead seek to account for possible uncertainties in our downstream analysis, for example by only computing a RoHO score for recurrent mutations for which we have at least three replicates (parent nodes) and at least two of each offspring carrying the reference and homoplastic allele.”

Indeed, our approach to handle phylogenetic uncertainty through filtering was commended by reviewer 3.

Consistently, other large-scale genomic studies of SARS-CoV-2 (Volz et al. 2020 bioRxiv; Phelan et al. bioRxiv 2020; Lu et al 2020. Cell, Worobey et al 2020 Science) do not employ tree resampling methods for assessing phylogenetic uncertainty for the same reasons eg. Lu et al 2020 Cell: “A phylogenetic bootstrap analysis was not performed due to the low number of phylogenetically informative sites”.

We have now added one further sentence to this effect in the manuscript:

*“In addition, it is of note that the SARS-CoV-2 population has only acquired moderate genetic diversity since its jump into the human population and, consequently, most branches in the phylogenetic tree are only supported by very few mutations. As a result of the low genetic diversity, most nodes in the tree have only low statistical support [41]. **Given boot-strapping analyses are not appropriate in this case (Lu et al 2020)**, we applied a series of stringent filters and masking strategies to the alignment (see Methods). Also, while our method does not account quantitatively for phylogenetic uncertainty, we only computed RoHO scores for situations which should be phylogenetically robust (i.e. mutations represented in at least three replicate nodes, each with at least two representatives of the reference and alternate allele in descendants).”*

and reiterated our strategy once more in the methods:

*“The latter allows us to consider only nodes for which we have multiple supported observations within the phylogeny, **conveniently accounting for phylogenetic uncertainty.**”*

It is computationally intractable to generate a set of posterior trees using MrBayes for an alignment of nearly 50 thousand genomes.

2. They have effectively ignored the point that homoplasies can be generate by recombination and that is undermines their analysis: a homoplasmy generated by recombination is obviously (!) not a recurrent mutation and it's appearance would mean there is no evidence for a selective advantage.

We very clearly mention the potential for recombination to introduce homoplasies (introduction paragraph 2) in our original submission.

We also stated in our response to this reviewer:

“We believe our approach to assessing whether a mutation is associated to differences in transmission is in fact largely immune to moderate levels of recombination since it does not matter whether (homoplastic) mutations were introduced in given lineages by de novo mutation or recombination.”

We purposely test all sites identified as homoplastic for signals of selection. As sites are tested independently it does not matter how they arise with any mutation, whether introduced by recombination or *de novo* mutation, ultimately having the potential to confer a selective advantage. We have added two statements to this effect in the methods:

"This avoids us taking forward homoplasies which have only been identified in a single geographic location as well as those putatively arising from low levels of recombination."

*"Under random sampling we expect that **any** mutation, **irrespective of how it arrives**, that positively affects a pathogen's transmission fitness will be represented in proportionally more descendant nodes."*

However, this is rather a moot point, as to reiterate there is currently no evidence for wide-spread recombination in the SARS-CoV-2 population. While this has been shown previously e.g. <https://observablehq.com/@spond/linkage-disequilibrium-in-sars-cov-2> we have also conducted an equivalent, and larger-scale, analysis on our dataset. We have detailed our approach to testing for recombination as a new section in the methods and added two new supplementary figures (Figure S6-S7):

"In order to test for the presence of recombination, we performed a linkage disequilibrium analysis considering whether the correlation between SNPs tends to disappear with an increase in the distance separating them on the genome. A classical approach to do so is the use of LD decay curves, that represent linkage disequilibrium as a function of the distance separating each SNP pair. We calculated metrics of linkage disequilibrium (r^2 and D') across all pair-wise combinations of variant sites using Tomahawk 0.7.0 (<https://github.com/mklarqvist/tomahawk>). The relationship between linkage disequilibrium and distance yielded a regression coefficient of $2.96e-08$ and the proportion of variance explained of $6.98e-5$ (Figure S6). Following the approach presented at <https://observablehq.com/@spond/linkage-disequilibrium-in-sars-cov-2> (accessed 21/09/2020) we tested the significance of the fitted r^2 by performing 1000 permutations of the genome coordinates, recomputing the regression each time. In all cases the observed values fell within the null distribution providing no evidence of recombination in the SARS-CoV-2 alignment (Figure S7)."

We reiterate this point clearly in the results:

"However, recurrent mutations **may be detected as a result of recombination**, for which we find no evidence in SARS-CoV-2 (Figure S6-S7), or sequencing or genome assembly artefacts [30]."

and discussion:

*" However, it is nonetheless interesting to consider the cause of these mutations. **Consistent with equivalent analyses** (<https://observablehq.com/@spond/linkage-disequilibrium-in-sars-cov-2>, accessed 21 September 2020), we find no signature of recombination in SARS-CoV-2 (Figures S6-S7)."*

3. The D614G mutation story is more complex than they say. The Volz paper in fact provides consistent evidence for a selective advance in this mutation. Hence, Van Dorp et al. are misrepresenting it. This highlights a major limitation of the whole Van Dorp study: it is not just recurrent mutations that will increase fitness. Indeed, a selective sweep on a single mutation is just as important as D614G shows. Hence, the whole focus on recurrent mutations is very limiting.

This statement is factually incorrect. Quoting directly from the abstract of Volz et al: "*Despite the availability of a very large data set, well represented by both Spike 614 variants, not all approaches showed a conclusive signal of higher transmission rate for 614G...*" which is completely consistent with our statement in the main text: "*A recent study on a sample of 25,000 whole genome sequences exclusively from the UK used different approaches to investigate D614G. Not all analyses found a conclusive signal for D614G, and effects on transmission, when detected, appeared relatively moderate.*"

Reviewers' Comments:

Reviewer #4:

Remarks to the Author:

This study looks recurrent mutations across a large phylogeny of SARS-CoV-2 and tests whether these mutations are linked to the relative number of descendants. Essentially this test seeks to identify any mutations leading to increased transmissibility in this virus. The authors propose a new statistic (Ratio of Homoplastic Offspring, or RoHO index) for this purpose.

The assumption that recurrent mutations are the best candidates for putative adaptation is plausible, but mutational hotspots will also produce similar patterns. Regardless, the authors obtain a negative result so this should not affect the conclusions. Recombination can also produce similar patterns. However, the authors demonstrate that there is no evidence of recombination in their dataset.

I did not see the earlier versions or the online preprint of this manuscript, so I have reviewed this paper on the latest version only. I have no major concerns about the analyses or interpretation of results, but I comment on three points raised by a previous reviewer of this submission.

1) Phylogenetic uncertainty

The analyses are based on a single estimate of the tree and do not account for phylogenetic uncertainty. Integrating the analysis over a set of trees (such as a set of trees from bootstrap replicates or a sample from a Bayesian posterior) would be more satisfying. As the authors point out in their rebuttal letter, a Bayesian analysis is not feasible for this dataset. The authors have taken reasonable steps to address phylogenetic uncertainty in their downstream analyses. Contrary to the statement on Line 301, bootstrapping would be a good way to generate a set of trees that could be used to account for phylogenetic uncertainty, even if it would not be useful for quantifying branch support.

2) Apparent homoplasies caused by recombination

The authors test for recombination and have not found any evidence for this in their dataset. I find these analyses convincing.

3) Limitations of looking at recurrent mutations

Mutations under strong positive selection (e.g. for increased transmissibility) may be missed by the approach used in this study, if they have experienced a selective sweep. However, this occurrence is probably unlikely because of the rapid population expansion of SARS-CoV-2. Even so, the authors should make the limitations of their approach clearer in the manuscript. The title of the manuscript specifically mentions 'recurrent mutations', so the authors have not overstated the findings of their study.

Additional minor comments

Line 49 'COVID-19' should be in upper case here and elsewhere

Line 51 'Since then, the virus has gradually accumulated mutations leading to patterns of genomic diversity' This statement doesn't seem very meaningful, as it is self-evident and universally true

Line 72 Why are most mutations expected to be neutral in SARS-CoV-2? Note also that whether mutations behave neutrally or not depends on the population size

Line 83 Why would the frequencies of neutral mutations increase through demographic processes?

Line 236 The first paragraph of the Discussion does not add much, and should be revised/condensed to reduce repetition

Thank you for the time taken by the editor to obtain a fourth reviewer. We feel the reviewer has fairly assessed our work and very much look forward to our findings being disseminated via Nature Communications.

We address Reviewer 4's queries below and have made amendments in the main text in line with their comments along with all editorial formatting suggestions. These are highlighted in yellow.

In addition, we have slightly amended Supplementary Figures S6 and S7. The results conveyed by these figures remain the same.

Reviewer #4 (Remarks to the Author):

This study looks recurrent mutations across a large phylogeny of SARS-CoV-2 and tests whether these mutations are linked to the relative number of descendants. Essentially this test seeks to identify any mutations leading to increased transmissibility in this virus. The authors propose a new statistic (Ratio of Homoplastic Offspring, or RoHO index) for this purpose.

The assumption that recurrent mutations are the best candidates for putative adaptation is plausible, but mutational hotspots will also produce similar patterns. Regardless, the authors obtain a negative result so this should not affect the conclusions. Recombination can also produce similar patterns. However, the authors demonstrate that there is no evidence of recombination in their dataset.

I did not see the earlier versions or the online preprint of this manuscript, so I have reviewed this paper on the latest version only. I have no major concerns about the analyses or interpretation of results, but I comment on three points raised by a previous reviewer of this submission.

1) Phylogenetic uncertainty

The analyses are based on a single estimate of the tree and do not account for phylogenetic uncertainty. Integrating the analysis over a set of trees (such as a set of trees from bootstrap replicates or a sample from a Bayesian posterior) would be more satisfying. As the authors point out in their rebuttal letter, a Bayesian analysis is not feasible for this dataset. The authors have taken reasonable steps to address phylogenetic uncertainty in their downstream analyses. Contrary to the statement on Line 301, bootstrapping would be a good way to generate a set of trees that could be used to account for phylogenetic uncertainty, even if it would not be useful for quantifying branch support.

Thank you for your positive assessment of how we account for phylogenetic uncertainty. We have amended the sentence on line 301 in line with the reviewer's suggestion:

"As a result of the low genetic diversity, most nodes in the tree have only low statistical support [41]. We therefore apply a series of stringent filters and masking strategies to the alignment (see Methods)."

2) Apparent homoplasies caused by recombination

The authors test for recombination and have not found any evidence for this in their dataset. I find these analyses convincing.

Thank you. To date we can detect no compelling evidence of recombination in SARS-CoV-2.

3) Limitations of looking at recurrent mutations

Mutations under strong positive selection (e.g. for increased transmissibility) may be missed by the approach used in this study, if they have experienced a selective sweep. However, this occurrence is probably unlikely because of the rapid population expansion of SARS-CoV-2. Even so, the authors should make the limitations of their approach clearer in the manuscript. The title of the manuscript specifically mentions 'recurrent mutations', so the authors have not overstated the findings of their study.

We have reiterated 'recurrent' when commenting on mutations in the Discussion section of the manuscript. As the reviewer states, we feel we have appropriately caveated our findings e.g. *"While such a method is obviously restricted to such recurrent mutations, it reduces the effect of demographic confounding problems such as founder bias"*.

Additional minor comments

Line 49 'COVID-19' should be in upper case here and elsewhere

We have amended this throughout.

Line 51 'Since then, the virus has gradually accumulated mutations leading to patterns of genomic diversity' This statement doesn't seem very meaningful, as it is self-evident and universally true

While self-evident to the research community we feel this is worth stating given the wider public interest of the manuscript.

Line 72 Why are most mutations expected to be neutral in SARS-CoV-2? Note also that whether mutations behave neutrally or not depends on the population size

We state this in line with the neutral theory of molecular evolution and cite Kimura & Ohta 1971 and consider this as the expectation when testing for sites under positive selection. We have added a clarifier to this sentence:

"While population genetics theory states that the majority of mutations are expected to be neutral [14], some may be advantageous or deleterious to the virus."

Line 83 Why would the frequencies of neutral mutations increase through demographic processes?

By demographic processes we refer to random genetic drift. We comment on founder bias later in the manuscript.

Line 236 The first paragraph of the Discussion does not add much, and should be revised/condensed to reduce repetition

We consider it important to restate our findings early in the discussion. Though, we removed from the first paragraph the sentence below as the point is already stated clearly in the results section.

"Although SARS-CoV-2 at present is effectively a single lineage with limited genetic diversity within it, the gradual accumulation of mutations in viral genomes in circulation may offer early clues to adaptation to its novel human host."